Fast and robust estimate of bacterial genus novelty using the percentage of conserved proteins with unique matches (POCPu)

Pauvert Charlie cpauvert@ukaachen.de
Hitch Thomas C.A.
Clavel Thomas tclavel@ukaachen.de
Functional Microbiome Research Group, Institute of Medical Microbiology, University Hospital of RWTH Aachen , Aachen , Germany
Wang Liang
Electronic publication date: 2025 Nov 14
Publication date: 2025
Volume: 13
Electronic Location ID: e20259
Received 2025 Aug 4; Accepted 2025 Sep 29
Copyright: ©2025 Pauvert et al.
Copyright year: 2025
Copyright holder: Pauvert et al.
License: This is an open access article distributed under the terms of the Creative Commons Attribution License, which permits unrestricted use, distribution, reproduction and adaptation in any medium and for any purpose provided that it is properly attributed. For attribution, the original author(s), title, publication source (PeerJ) and either DOI or URL of the article must be cited.
License URL: https://creativecommons.org/licenses/by/4.0/

Keywords: Bacterial taxonomy, Bacterial genomics, Genus delineation, Protein sequence comparison, Percentage of conserved proteins (POCP), Benchmarking

Funding: German Research Foundation (DFG) 460129525 (NFDI4Microbiota) 445552570 Thomas Clavel received funding from the German Research Foundation (DFG): project no. 460129525 (NFDI4Microbiota), and project no. 445552570. The funders had no role in study design, data collection and analysis, decision to publish, or preparation of the manuscript.

==============================
Accurate taxonomic assignment of bacterial genomes is essential for identifying novel taxa and for stable classification to enable robust comparison between studies. Bacterial genus delineation relies on multiple lines of evidence, including phylogenetic trees and metrics like the percentage of conserved proteins (POCP). POCP is widely used, but requires benchmarking in terms of both, computation and accuracy. We used 2,358,466 pairwise comparisons of proteomes derived from 4,767 genomes across 35 families to systematically assess POCP calculation and percentage of conserved proteins with unique matches (POCPu) which considers unique matches only. Both methods are 20x faster than the reference BLASTP when using the very-sensitive setting of DIAMOND. However, POCPu differentiates better within-genus from between-genera values, which improves bacterial genus assignment. This work facilitates comparative analysis of an increasingly larger number of genomes, providing a reliable metric to support genus delineation. The findings suggest that specific POCPu thresholds deviating from the reference 50% value are needed for certain families.

Introduction

Bacterial taxonomy is the classification of strains into lineages ranging from phyla to species (Tindall et al., 2010). This is critical to understand microbial diversity by creating a coherent framework that reflect their evolutionary relationships. Two elements show that accurate taxonomic placement of microorganisms is more important now than ever: (i) a very high fraction of bacteria, both in the environment and host-associated microbiomes, remain to be described and named (Thomas & Segata, 2019; Sutcliffe, Rosselló-Móra & Trujillo, 2021; Rodríguez del Río et al., 2024); (ii) large-scale metagenomic studies in the last decade and now high-throughput cultivation methods are accelerating the pace of bacterial discovery (Hugenholtz et al., 2021; Clavel et al., 2025). It is therefore essential to consolidate the system for classifying bacteria and confidently assess their taxonomic novelty. It is equally important to ensure that the boundaries between genera that include known taxa are stable. Whilst bacteria had been classified based on morphology and phenotypic parameters for decades, the advent of genomics has revolutionised the way we classify bacteria (Chun & Rainey, 2014), giving rise to several overall genome relatedness indices (OGRI). However, in the process of classifying bacteria, it is essential to provide a robust framework that includes easy-to-implement parameters to classify organisms at each level of bacterial lineage, including the genus level, which is the focus in this article.

The Genome Taxonomy Database (GTDB) has proposed a standardized approach to group publicly available genomes into species clusters (Parks et al., 2020), resulting in an invaluable resource that is regularly updated and adopted by the community (Parks et al., 2022). The approach is resilient to genome contamination, which can plague public repositories (Mussig et al., 2024), but also includes cases of taxonomic incongruence with previously described and accepted species names (Parks et al., 2020).

In the last three years alone, the number of bacterial genomes just in the RefSeq collection, a high-quality and curated subset of public sequence databases, has increased by 35,000 per year, both from isolates and metagenomes (Haft et al., 2024). To analyse these genomes, we need clear and rapid methods for taxonomic assignment. For species, the average nucleotide identity (ANI) has been developed (Jain et al., 2018) and shown to delineate species almost unanimously (Parks et al., 2020). Whilst there is no ANI threshold for genus delineation (Qin et al., 2014), family-specific thresholds have been suggested using both the ANI value and alignment fraction, however the lack of a clear threshold limits their usability (Barco et al., 2020). An alternative to ANI is the average amino acid identity (AAI) which uses protein sequences instead of genomic nucleic sequences (Konstantinidis & Tiedje, 2005). Whilst several tools implement AAI calculation (Medlar, Törönen & Holm, 2018; Kim, Park & Chun, 2021; Dieckmann et al., 2021; Gerhardt et al., 2025), and AAI values above 65 to 95% were proposed for genomes from the same genus (Konstantinidis, Rosselló-Móra & Amann, 2017), such wide range requires to combine AAI with others metrics to classify genera. A protein sequence-based genus delineation method with an interpretable metric is the percentage of conserved proteins (POCP) (Qin et al., 2014). If two bacterial genomes share more than half of their conserved proteins, i.e., POCP >50%, they are considered to represent species from the same genus.

POCP is widely used, in combination with other OGRI, to assign novel bacterial taxa to known genera, or to support the proposal of novel genera (Orata et al., 2018; Chaplin et al., 2020; González et al., 2020; Wylensek et al., 2020; Liu et al., 2021; Kuzmanović et al., 2022; Afrizal et al., 2022; Sereika et al., 2023; Hitch et al., 2025). However, the validity of the reference threshold value (50%) has not been widely tested. In addition, a major limitation of POCP is that comparing all proteins within each genome to each other is computationally demanding. Given that the number of valid genus names almost doubled (Fig. S1) since the original proposition of POCP by Qin et al. (2014), scalable methods and a timely re-evaluation of the POCP approach are needed.

Hernández-Salmerón & Moreno-Hagelsieb (2020) compared protein alignment tools to find faster alternatives to identify reciprocal best hits, without a loss in precision. They found that DIAMOND (Buchfink, Reuter & Drost, 2021), set to sensitive parameters instead of defaults, correctly found 87% of the reciprocal best hits of BLASTP (Camacho et al., 2009) in less than 8% of the computing time. Recently, Hölzer (2024) suggested the use of DIAMOND with ultra-sensitive settings to compute POCP faster than with BLASTP. This method was based on previously available code (Hölzer, 2020), implemented as a nextflow workflow (Hölzer, 2024). However, this new method was only validated on five genera, with 15 to 167 genomes each. Fundamental changes, such as the tool selected, can have significant effects on calculations, especially if 13% (as in 100%–87%) of matching proteins may not be found. Current implementations of POCP have also modified the calculation method by considering conserved proteins to unique matches (Hölzer, 2020; Hölzer, 2024; Lin, 2021; Riesco & Trujillo, 2024), without comparing to the original implementation (Qin et al., 2014). These studies clearly show the need for a clear definition of POCP to avoid divergent assumptions in tools between microbiologists and developers. Furthermore, given the increased number of genomic resources available, we need a fast and reproducible framework to classify genera that needs to be tested at a large scale.

Here the aim was to re-evaluate genus-level delineation based on POCP proposed 10 years ago, with a focus on scalability and the handling of duplicate genes. To achieve this, we analyzed 2,358,466 pairwise comparisons of 4,767 genomes across 35 families, with a focus on optimizing POCP implementation to provide a reliable and accurate metric that can be used in conjunction with other evidence to support genus delineation.

Materials and Methods

Standardisation of protein sequences and taxonomy via GTDB

Portions of this text were previously published as part of a preprint (bioRxiv https://doi.org/10.1101/2025.03.17.643616). As GTDB provides curated taxonomy along with genomes and genome-derived protein sequences (Parks et al., 2022), we used it as a reliable source of high-quality data in our benchmark. We used inclusion criteria to facilitate the selection of a diverse range of taxonomic groups from GTDB (release 214) (N = 394,932 bacterial genomes), while maintaining achievable comparisons with the time, human, and computing resources available: (1) the bacteria had a valid name according to the list of prokaryotic names with standing in nomenclature (Parte et al., 2020) and a representative genome was available (N = 11,699), (2) they belonged to a family with at least two genera (N = 5,904), and (3) to a genus with at least ten genomes (N = 4,767). Based on these criteria, the protein sequence files for the shortlisted bacteria were obtained from GTDB (Table S1) which uses Prodigal v2.6.3 (Hyatt et al., 2010) for protein sequence prediction. A single representative genome from each species, as designated by GTDB, was selected for further analysis (Table S1).

Definition of percentage of conserved proteins (POCP)

The percentage of conserved proteins (Fig. 1A) between two genomes Q and S is defined as: (1) POCP=CQS+CSQTQ+TS×100%

where CQS represents the conserved number of proteins from Q when aligned to S and conversely CSQ represents the conserved number of proteins from S when aligned to Q; TQ + TS represents the total number of proteins in the two genomes being compared (adapted from Qin et al. 2014). The range of POCP is theoretically [0; 100%]. Conserved proteins are defined as protein sequences matches from the query with an e-value <10−5, a sequence identity >40%, and an aligned region >50% of the query protein sequence length (Qin et al., 2014).

Figure 1 Schematic overview of POCP and POCPu calculation and the overall benchmarking workflow.

(A) Three simplified examples of bacterial genomes (two from the same genus and one from a different genus) to illustrate how the percentage of conserved proteins (POCP) or percentage of conserved proteins using unique matches (POCPu) were calculated. Details on the formula for POCP and POCPu calculation are provided in the Methods. (B) Overview of the benchmarking workflow. Shortlisted, high-quality genomes from the Genome Taxonomy Database (GTDB) were used to compare different protein alignment methods (top-right) and to evaluate the potential of POCP and POCPu for genus delineation (bottom-right).

Definition of Percentage of Conserved Proteins using only unique matches (POCPu)

During the alignment process, protein sequences from the query can match multiple subject sequences in the case of duplicated genes (Fig. 1A). Whilst briefly mentioned in the original article that “The number of conserved proteins in each genome of strains being compared was slightly different because of the existence of duplicate genes (paralogs)” (Qin et al., 2014, p. 2211), the impact of this on POCP values was not determined. Therefore, we defined the POCP with unique matches (POCPu) between two genomes Q and S as: (2) POCPu=CuQS+CuSQTQ+TS×100%

where CuQS represents the conserved number of proteins from the unique matches of Q when aligned to S and, conversely, CuSQ the conserved number of proteins from the unique matches of S when aligned to Q; TQ + TS represents the total number of proteins in the two genomes being compared. We hypothesized that the impact of paralogs on POCP will decrease with unique matches as only one pair of duplicated conserved genes will be counted instead of two (Fig. 1A).

Pairwise comparisons between a query sequence (Q) and a subject sequence (S) were defined by banning self-comparisons (Q ≠ S) and considering reciprocal comparisons (Q − S and S − Q) only within the same family to avoid unnecessary expansion of the comparison landscape.

Finding suitable protein sequence alignment methods to scale BLASTP-based POCP calculations

In Qin et al. (2014), guidance was provided on how to implement the computation of POCP, including the use of BLASTP. As ‘standard’ POCP method, we used BLASTP v2.14.0+ (Camacho et al., 2009) with parameters from Qin et al. (2014) (Table S2). We also considered a modified implementation of the BLASTP method, named BLASTPDB where BLAST databases are first built for the two genomes considered (Table S2). This allows parallel alignments on multiple CPU, which is not possible with BLASTP. We then included two tools that were designed as faster local-protein-alignment methods and used as alternatives to BLASTP: DIAMOND v2.1.6 (Buchfink, Reuter & Drost, 2021) and MMseqs2 v15.6f452 (Steinegger & Söding, 2017). The former is used in Hölzer (2024) while the latter is used in EzAAI (Kim, Park & Chun, 2021). Similar to BLASTPDB, these methods require that a protein database is built for each genome before performing the alignment (Table S2). DIAMOND and MMseqs2 were both used with four different sensitivity thresholds proposed recently (Table S2) (Buchfink, Reuter & Drost, 2021). This benchmark (Fig. 1B) was run on a subset of 1,235 GTDB genomes. We used genomes from a wide range of bacterial phylogenetic diversity across four phyla: Bacillota (6 families, 23 genera, 561 species), Pseudomonadota (6 families, 16 genera, 333 species), Bacteroidota (3 families, 7 genera, 217 species) and Actinomycetota (2 families, 6 genera, 124 species).

All protein matches were filtered to only keep matches with >40% identity to all the query sequences matches for POCP (CQS and CSQ in Eq. (1)) and only unique query sequence matches for POCPu (CQS and CSQ in Eq. (2)). The filtering was adapted to the method as the range of percentage of identity in MMseqs2 is [0 − 1] and [0 − 100] for BLAST and DIAMOND. The total number of proteins per genomes (TQ and TS in Eqs. (1) and (2)) was computed using seqkit stats v2.2.0 (Shen et al., 2016). Linear regressions were implemented using R version 4.3.1 (2023-06-16) to fit the expected POCP (or POCPu) values obtained via the BLASTP reference method against the other methods considered. The coefficient of determination R2 of the linear regression is used as an interpretable and bounded goodness-of-fit measure between the expected and measured values instead of other measurement errors (Chicco, Warrens & Jurman, 2021). We did not rely on the adjusted coefficient of determination as the linear regressions had only one predictor, namely the POCP (or POCPu) values of the evaluated method.

Evaluate POCP and POCPu with known within-genus versus between-genera pairs

This analysis was run as described above except that only the scalable approach to compute POCP (or POCPu) was used on the full set of genomes (N = 4,767) to capture additional diversity (Fig. 1B): Pseudomonadota (15 families, 66 genera, 1,736 species), Actinomycetota (eight families, 27 genera, 1,584 species), Bacteroidota (six families, 27 genera, 886 species) and Bacillota (six families, 23 genera, 561 species). Next, we computed classification metrics with a positive event defined as “both genomes belong to the same genus”. Thus, for a pair of bacterial genomes with a POCP (or POCPu) >50%, the pair was a true positive (TP) if it belonged to the same genus, else it was considered as a false positive (FP). Conversely, for a pair of bacterial genomes with a POCP (or POCPu) ≤ 50%, the pair was a false negative (FN) if it belonged to the same genus, else it was considered as a true negative (TN). We then assessed the classification performance of both POCP and POCPu using Matthews correlation coefficient (MCC; Eq. (3)). (3) MCC=TP×TN−FP×FNTP+FPTP+FNTN+FPTN+FN.

The MCC coefficient ranges from −1 to 1 and is high in the case of a perfect classification, whilst a value of 0 indicates random classification. Negative MCC values indicate perfect misclassification, as in a swap between positive and negative events. In addition, the MCC compensates for imbalanced datasets compared to other metrics such as accuracy or F1-score (Chicco & Jurman, 2020). Finally, we used one dimensional optimization (Brent, 1972) to find family-specific POCPu thresholds maximizing MCC values and separating between-genera from within-genera distributions. The optimization was run using the optimize () function from the stats package (R Core Team, 2023).

Benchmarking workflow implementation

Automatic protein sequences download, data pre-processing, many-versus-many protein alignments, POCP computation and delineation metrics calculations have been included in a workflow using nextflow v23.10.0 (Di Tommaso et al., 2017), based on components of nf-core (Ewels et al., 2020). The tools used are provided within Docker container (Merkel, 2014) or bioconda (The Bioconda Team et al., 2018) environments to ensure reproducibility and scalability and to facilitate future extension of the present benchmarking work. Nextflow natively keeps track of the time, CPU, memory, and disk usage of each process in an execution trace log file, which we used to evaluate the computing resources utilization. Process duration is available as wall-time and real-time, the CPU usage is reported as a percentage of usage of a unique CPU, meaning multi-threaded processes will have a value higher than 100%. Statistical analyses and visualization were conducted in R using targets v.1.7.0 (Landau, 2021).

Results

Finding an alternative to BLASTP-based POCP calculations

First, we set out to identify a scalable alternative to BLASTP to compute the POCP to delineate genera (Fig. 1). We evaluated ten protein alignment methods (Table S2) based on three tools—BLASTP (Camacho et al., 2009), DIAMOND (Buchfink, Reuter & Drost, 2021) and MMseqs2 (Steinegger & Söding, 2017).

We processed 1,235 genomes and conducted 1,412,040 pairwise comparisons with a total of 32,316 CPU-Hours (3.7 in years). All the methods tested were faster than the reference BLASTP (Table S3). BLASTPDB, the database method of BLASTP (Table S2) that enables paralleled computations, was only half the time of BLASTP on average. In contrast, all DIAMOND and MMSEQS2-based methods ran at 20x and 11x the speed of BLASTP, respectively at the cost of using more memory, CPU, and disk usage (Table S3). Thus, as expected, more sensitive methods consumed more resources in general. An important criterion for a BLASTP alternative is to ensure that the increased speed does not compromise the accuracy of POCP calculation.

DIAMOND provides POCP values as accurate as with BLASTP

BLASTPDB produced the exact same POCP values as BLASTP (Fig. S2). The other methods did not perform as good. All methods of DIAMOND had a coefficient of determination (R2) above 0.99, except for DIAMOND_FAST that deviated from the expected values (Fig. 2).

Figure 2 Adequacy between POCP values computed with the reference method BLASTP and methods of faster alternatives: DIAMOND (Buchfink, Reuter & Drost, 2021) and MMSEQS2 (Steinegger & Söding, 2017).

Each point (n = 70,602 per tool) represents a POCP value between two genomes (see Eq. (1)). The colors represent the number of data points binned together in hexagons to avoid over-plotting. Coefficient of determination (R 2) and associated p-value are shown on top of each linear regressions.

All DIAMOND methods, especially DIAMOND_FAST, tended to underestimate POCP values (all dots were below the reference dashed line), meaning that they might assign genomes to different genera when they are from the same genus (top panels). Deviation from the BLASTP reference was aggravated when using the MMSEQS2_S1DOT0 method, mainly through underestimation of POCP values (bottom panels). The other MMSEQS2 methods performed better, but still less good than the DIAMOND methods. They also tended to overestimate more than underestimate. All in all, the DIAMOND methods, especially with increased sensitivity, generated POCP values nearly as accurate as BLASTP for a fraction of the time, but we refrained from using the MMSEQS2 methods due to being less accurate.

Proposal for clear and fast computation of POCP values

We observed that all methods generated POCP values exceeding the supposed upper limit of 100%. Hence, we investigated the underlying reasons and provide a clearer definition of POCP, termed POCPu (see Eq. (2)).

POCP values above 100% disappeared when using POCPu (Fig. 3 and Fig. S3). In general, the same patterns observed for POCP hold for POCPu, though with higher values of coefficient of determination (Fig. 3 and Table 1). The three different sensitive methods of DIAMOND produced POCPu values that matched perfectly the ones produced by the reference method BLASTP, with no underestimation as in the case of POCP. In contrast, the MMSEQS2 methods, whilst better with POCPu than POCP, still tended to underestimate POCPu values. All in all, DIAMOND-based POCPu is closer to its reference than POCP (Fig. 3), guiding our choice to create an accurate BLASTP alternative.

Figure 3 Adequacy between POCPu values computed with the reference method BLASTP and methods of faster alternatives: DIAMOND (Buchfink, Reuter & Drost, 2021) and MMSEQS2 (Steinegger & Söding, 2017).

Each point (n = 70,602 per tool) represents a POCPu value between two genomes (see Eq. (2)). The colors represent the number of data points binned together in hexagons to avoid over-plotting. Coefficient of determination (R 2) and associated p-value are shown on top of each linear regressions.

Table 1 POCP and POCPu linear regressions results for each alternative method to BLASTP.

Method	POCP	POCPu	
	R 2	p-value	R 2	p-value	
BLASTPDB	1.0000000	<0.001	1.0000000	<0.001	
DIAMOND_ULTRASENSITIVE	0.9920389	<0.001	0.9997605	<0.001	
DIAMOND_VERYSENSITIVE	0.9919744	<0.001	0.9997574	<0.001	
DIAMOND_SENSITIVE	0.9916398	<0.001	0.9997438	<0.001	
DIAMOND_FAST	0.8967444	<0.001	0.9913959	<0.001	
MMSEQS2_S7DOT5	0.9379505	<0.001	0.9722865	<0.001	
MMSEQS2_S6DOT0	0.9369705	<0.001	0.9720813	<0.001	
MMSEQS2_S2DOT5	0.9242783	<0.001	0.9713977	<0.001	
MMSEQS2_S1DOT0	0.8659630	<0.001	0.9713360	<0.001	
Notes.

Coefficient of determination (R2) and associated p-value for linear regressions matching the POCP and POCPu values computed by each method against the respective POCP and POCPu values of the reference method BLASTP. Each linear regression are based on n = 70, 602 comparisons per method. Methods are sorted by decreasing POCPu values.

In summary, the BLASTPDB method performed as good as BLASTP (Figs. S2, S3 and Table 1) in half the time (Table S3), at the cost of using more resources. However, the DIAMOND sensitive methods were even faster with excellent adequacy with BLASTP, especially for POCPu (Table 1). Whilst DIAMOND_ULTRASENSITIVE had the highest R2 value using POCPu (Table 1), it also had the highest memory consumption and disk usage (Table S3). A more sustainable alternative is DIAMOND_VERYSENSITIVE that performed 10 times faster than BLASTPDB, in less than 5% of the time of BLASTP, while still maintaining reasonable usage of the computing resources (Table S3). Importantly, POCPu values calculated using DIAMOND_VERYSENSITIVE delivered results extremely close to the reference BLASTP (Fig. 3 and Table 1) and were essentially identical to DIAMOND_ULTRASENSITIVE POCPu R2 up to 5 digits (Table 1). Therefore, we consider DIAMOND_VERYSENSITIVE to be a valid and scalable alternative to BLASTP for POCP/POCPu computations.

Evaluate POCP and POCPu using within- and between-genera pairs

Unique matches enhance the accuracy of genus delineation

Next, we evaluated the 50%-threshold of POCP and POCPu to determine their reliability to delineate bacterial genera. We included all 4,767 genomes (Fig. 1) and calculated POCP and POCPu for 1,087,630 pairwise comparisons using DIAMOND_VERYSENSITIVE (Fig. 4).

Figure 4 Genus delineation with the reference 50% value.

Distribution of POCP (A) and POCPu (B) values for all pairwise genome comparisons: Between-genera (n = 321,189) in orange and Within-genus (n = 222,626) in sky blue. The GTDB taxonomy was used as reference for the confusion matrix (true/false positives and negatives). POCP and POCPu values were calculated with our recommended method DIAMOND_VERYSENSITIVE (Table S2); they range from 20 to 236.9 for POCP and 16.9 to 94.6 for POCPu. The dashed lines indicate the standard 50% threshold for genus delineation.

Instead of two bell-shaped distributions, with the 50% threshold separating between-genera (left) from within-genus POCP values (right), we observed overlapping POCP distributions (Fig. 4A). This was associated with a high number of false positives (FP = 188,155), where between-genera values were >50%, especially compared with the number of true negatives (TN = 133,034), where between-genera values were <50%. Additionally, most of the within-genus values were >50% (TP = 220,307), with only few below the threshold (FN = 2,319).

In contrast, POCPu was much closer to the expected results given the taxonomic assignments of each genome (Fig. 4B). Between-genera POCPu values followed a bimodal distribution, with the highest peak and most of the distribution remaining below the 50%-threshold (TN = 253,860). Nonetheless, a fraction of between-genera values were higher than the threshold of 50%, representing false positives, i.e., different genera when they are not (FP = 67,329). As in the case of POCP, within-genus POCPu values were above the threshold of 50% (TP = 209,660), with a few below the threshold (FN = 12,966). All in all, considering only unique protein matches improves genus delineation.

To quantify these findings on the confusion matrix (true/false positives and negatives), we used the MCC (Eq. (3) in the Methods), which is a binary classification rate that gives a high score only when the classifier correctly predicts most of positive and negative cases. POCPu (MCC = 0.72) surpassed POCP (MCC = 0.46) to delineate bacterial genera, which quantitatively confirmed the visual findings (Fig. 4).

Family-specific POCPu thresholds enable clearer genus delineation

Analysing the bacterial families separately questioned the universal threshold of 50% conserved proteins (Fig. 5 and Fig. S4). The large family of Streptomycetaceae (Actinomycetota, Fig. 5A), was characterized by a very low MCC, and thus many false cases, including FN = 2,974 (2.8%) and FP = 7,110 (6.8%), despite even more true case TN = 1,231 (1.2%) and TP = 93,338 (89.2%). In 7 families out of 35, POCPu was clearly not adequate to delineate genus using a threshold of 50%, as indicated by low MCC (MCC ≤ 0.25; Fig. 5B). In contrast, POCPu delineated bacterial genera accurately for 18 families (MCC ≥ 0.7; Fig. 5B).

Figure 5 POCPu delineates bacterial genera in a family-specific manner.

(A) Three representative examples of family-specific genus delineation capacity where POCPu values can (i) be neatly distinct and allow for genus delineation (top; example = Xanthobacteraceae); or (ii) overlap and hamper genus delineation (bottom; example = Streptomycetaceae); or (iii) any scenario in between (middle; example = Lactobacillaceae). The dashed lines indicate the standard threshold of 50% conserved proteins. (B) The ability of POCPu to delineate genera was quantified for each of the 35 families analysed using the Matthews correlation coefficient (MCC, Chicco & Jurman, 2020). An MCC of −1 and +1 indicates perfect misclassification or classification, respectively; random genus delineation corresponds to MCC = 0. The dashed line indicates the global MCC on the whole dataset, across all families. The number of genomes included per family are indicated in brackets next to the family names. The families were ranked per phyla (alphabetically; vertical facets in grey) and then by decreasing MCC within each phylum. The visual POCPu distributions for all families are provided in Fig. S4.

Due to these differences between families, we explored family-specific POCPu thresholds by maximizing MCC to improve genus delineation (Fig. S4 and Table 2). With this procedure, thresholds other than the default 50% would enhance classification for 19 families out of the 35 families. The genus delineation of eight families was improved, with at least 0.1-point increase in MCC (white squares in Table 2; Actinomycetota: Streptomycetaceae and Streptosporangiaceae; Bacillota: Amphibacillaceae, Lactobacillaceae and Metamycoplasmataceae; Pseudomonadota: Acetobacteraceae, Burkholderiaceae_B and Vibrionaceae). For 11 additional families maximum MCC above 0.7 were even obtained (black squares in Table 2; Actinomycetota: Micromonosporaceae and Pseudonocardiaceae; Bacteroidota: Bacteroidaceae and Weeksellaceae; Pseudomonadota: Burkholderiaceae, Enterobacteriaceae, Halomonadaceae, Pseudomonadaceae, Rhizobiaceae, Rhodobacteraceae and Xanthomonadaceae). Interestingly, in two cases, the optimal POCPu threshold was lower than the standard threshold: 43% for Bacteroidaceae (Bacteroidota) and 45.5% for Metamycoplasmataceae (Bacillota). In 17 cases, new thresholds higher than 50% conserved proteins better separated genomes from within genus and between genera (Table 2).

Table 2 Proposal of family-specific POCPu thresholds for genus delineation.

	Family	ΔMCC	Threshold value (%)		MCC	Max. MCC	
Actinomycetota	
	Mycobacteriaceae	0.02	52.3		0.89	0.91	
	Micrococcaceae	0.07	53.9		0.88	0.95	
	Nocardioidaceae	0.00	48.8		0.74	0.74	
□	Streptosporangiaceae	0.15	63.1	↑	0.72	0.87	
	Microbacteriaceae	0.10	53.7		0.63	0.73	
■	Pseudonocardiaceae	0.23	56.2	↑	0.63	0.86	
■	Micromonosporaceae	0.41	57.9	↑	0.57	0.98	
□	Streptomycetaceae	0.33	55.7	↑	0.16	0.49	
Bacillota	
	Planococcaceae	0.00	51.3		1.00	1.00	
	Streptococcaceae	0.03	47.7		0.93	0.97	
□	Metamycoplasmataceae	0.14	45.5	↓	0.81	0.95	
	Paenibacillaceae	0.06	48.3		0.75	0.81	
□	Amphibacillaceae	0.21	53.2	↑	0.74	0.95	
□	Lactobacillaceae	0.11	57.9	↑	0.71	0.82	
Bacteroidota	
	Hymenobacteraceae	0.01	53.6		0.99	1.00	
	Spirosomaceae	0.02	52.9		0.98	1.00	
	Sphingobacteriaceae	0.01	50.4		0.92	0.92	
■	Bacteroidaceae	0.17	43.0	↓	0.63	0.80	
	Flavobacteriaceae	0.01	51.4		0.59	0.60	
■	Weeksellaceae	0.63	60.1	↑	0.22	0.85	
Pseudomonadota	
	Xanthobacteraceae	0.00	45.8		1.00	1.00	
	Beijerinckiaceae	0.00	50.1		0.97	0.97	
	Moraxellaceae	0.02	53.2		0.92	0.94	
□	Burkholderiaceae_B	0.12	54.1	↑	0.85	0.97	
	Alteromonadaceae	0.01	49.3		0.78	0.79	
	Sphingomonadaceae	0.10	52.9		0.63	0.73	
■	Burkholderiaceae	0.22	64.3	↑	0.52	0.74	
■	Rhizobiaceae	0.43	62.5	↑	0.48	0.91	
■	Rhodobacteraceae	0.27	58.1	↑	0.43	0.70	
■	Xanthomonadaceae	0.51	63.7	↑	0.33	0.85	
□	Vibrionaceae	0.28	59.3	↑	0.23	0.51	
■	Enterobacteriaceae	0.60	71.2	↑	0.19	0.79	
■	Halomonadaceae	0.56	59.6	↑	0.19	0.76	
■	Pseudomonadaceae	0.82	63.2	↑	0.07	0.89	
□	Acetobacteraceae	0.63	64.3	↑	0.05	0.68	
Notes.

The thresholds were obtained after maximizing the MCC value to separate between-genera from within-genera distributions. Squares indicate that MCC value change was greater than 0.1 with the optimized threshold, filled squares denote rescued families from MCC < 0.7 to MCC > 0.7, whilst empty squares indicate improved genus delineation. Arrows highlight potential family-specific threshold worth considering to replace the default of 50% with the direction of change. Families without squares already delineate genera correctly with the default of 50%.

Because POCP was previously proposed to be influenced by genome size (Riesco & Trujillo, 2024), we used the large pairwise comparisons dataset to assess whether the changes in threshold were linked to differences in genome size. If POCPu is influenced, we reasoned that its genus delineation power should also be influenced, therefore we expected stronger genome size differences in the families for which an alternative POCPu threshold was found. We found no evidence that POCPu is affected by differences in genome size (Fig. S5A) nor proteins number (Fig. S5B).

Discussion

Qin et al. (2014) proposed to separate bacterial genera using the POCP in genomes more than 10 years ago. POCP is one of several commonly used metrics to delineate bacterial genera (others are AAI or 16S rRNA gene sequence identity), but it requires efficient and accurate calculation. The descriptions of many novel genera report POCP values oscillating around the proposed threshold value of 50%, suggesting it is not a clear-cut separation (Wylensek et al., 2020; Afrizal et al., 2022; Hitch et al., 2025). We therefore set out to re-evaluate genus-level delineation based on POCP using a comprehensive dataset, and to underline a faster and clearer method. We show that DIAMOND_VERYSENSITIVE can reliably replace BLASTP, speeding up the computing process by 20x. In addition, we addressed an assumption made in previous POCP implementations (Hölzer, 2020; Hölzer, 2024; Lin, 2021), and thereby clearly defined an alternative POCP metric–POCPu–that uses only unique matches, making genus delineation more accurate.

Genus names occur before species names in the binomial nomenclature of bacteria, and are therefore an important first contact with bacterial entities. They are key to existing knowledge in databases or articles and provide intuitive information on the evolutionary history and ecological roles of the organisms under study (Schoch et al., 2020; Reimer et al., 2022; Rosonovski et al., 2024). The system works best when resources and tools follow FAIR principles (Wilkinson et al., 2016; National Microbiome Data Collaborative, 2025; NFDI4Microbiota, 2025), as done in this work. In addition, our selected DIAMOND-based POPCu improves sustainability through faster computation and hence reduced electrical consumption, although the additional benefit of decreased disk usage is not captured by the carbon footprint estimator (Lannelongue, Grealey & Inouye, 2021). We demonstrated that not all tools and parameters are suitable to speed up BLASTP; some combinations, whilst extremely fast, under- or overestimate POCP values, resulting in erroneous splitting or merging of genera.

This study has several limitations. While our analysis included phylogenetically diverse taxa sourced from various environments, we might have missed important taxonomic groups of interest to readers (e.g., Lachnospiraceae or Clostridiaceae). However, filtering was necessary to obtain enough data points per taxa to ensure statistical robustness, and we kept enough genomes to represent a broad bacterial diversity far beyond the type species of genera. Previous studies on many-vs-many protein alignment comparison used less phylogenetic diversity: four genomes from four genera in Hernández-Salmerón & Moreno-Hagelsieb (2020), or up to 167 genomes from five genera in Hölzer (2024). Riesco & Trujillo (2024) evaluated much more genomes, 1,573, but they included only type strains of type species of genera and calculated POCP–using Bio-Py (Lin, 2021)–only to compare with AAI and not to evaluate genus delineation. Another limitation is that our study relies fully on GTDB as the source of genomes and taxonomy due to being a very comprehensive taxonomic resource and a reference worldwide (Parks et al., 2020; Parks et al., 2022). The simulation of genomes with defined mutation and recombination rates to mimic within-genus and between-genera populations could be a useful approach in future work. We also acknowledge that this work focused on benchmarking POCP; it will be interesting in the future to compare the accuracy of genus delineation using other OGRI such as AAI, ANI, or new enhanced approaches that include structural protein information to consider distant and functional homologs in POCP calculation.

Threshold-based approaches are always a matter of compromise, and do not provide one-size-fits-all solution. Regarding species delineation, Parks et al. (2022) stated: “The use of ANI to delineate species despite the lack of clear evidence for discrete species boundaries in the GTDB dataset is a pragmatic approach for organizing the rapidly growing biodiversity being discovered with metagenomic approaches”. We share their vision and propose the use of POCPu as an interpretable and pragmatic approach to delineate bacterial genera. In an effort to improve this process, we suggest refining the classification by applying family-specific POCPu thresholds, as shown previously for Rhizobiaceae (Kuzmanović et al., 2022). However, one should only deviate from the standard threshold of 50% if the benefit is greater than the risk of creating more confusion. We have provided tentative thresholds for several families, for which confidence was high. Indeed, initiatives like the SeqCode (Hedlund et al., 2022; Jiménez & Rosado, 2024) require reliable methods to assess taxonomic novelty, and we propose that POCPu is a robust yet scalable approach for modern taxonomy. Finally, it is important to remember that accurate taxonomic placement is best achieved when multiple lines of evidence are considered, as implemented in Protologger (Hitch et al., 2021). In the case of genera, POCPu decisions can be supported, for example by assessing the topology of phylogenetic trees, considering 16S rRNA gene identities (Yarza et al., 2014; Hackmann, 2025), and the result of GTDB-Tk analysis (Chaumeil et al., 2022).

Conclusions

Percentage of conserved proteins is a widely used index for genus-level delineation of bacteria but it requires benchmarking using large-scale genomic data. Here we provide an up-to-date method to identify homologous protein sequences. We optimized POCP using 2,358,466 pairwise comparisons of genomes from the GTDB. DIAMOND emerged as a faster yet accurate replacement for BLASTP when using very sensitive settings. Plus, we refined POCP using unique matches only (POCPu), which improved separating between-genera from within-genus distributions. Our benchmark enabled us to rapidly evaluate POCPu values on 143 bacterial genera across 35 families and four phyla, which highlighted specific POCPu thresholds around the reference 50% value for certain families. Overall, we consolidated one line of evidence in bacterial taxonomy with a fast and robust index that will strengthen bacterial genus delineation.

Supplemental Information

Supplemental Information 1 Shortlisted genomes of the Genomes Taxonomy DataBase (GTDB)

The genomes included passed all criteria for our benchmark: if (1) the bacteria had a valid name according to the List of Prokaryotic names with Standing in Nomenclature and a representative genome was available, (2) they belonged to a family with at least two genera, and (3) to a genus with at least ten genomes.

Supplemental Information 2 List of the ten methods and associated parameters for the many-versus-many proteins alignments tools used in the benchmark

The recommended approach is indicated in bold.

Supplemental Information 3 Fold change of computing metrics for the ten methods used in the benchmark compared to the BLASTP method

The metrics include processing time as real-time, memory usage, CPU usage and disk usage as input/output (I/O). A fold change below 1 means the metric was lower, above 1 means it was higher, compared to the reference. The fold change values are median computed over n = 141,204 number of processes tracked per approach.

Supplemental Information 4 Cumulative number of validly published genera names according to the International Code of Nomenclature of Prokaryotes (ICNP)

The year 2014 is highlighted as it corresponds to the year of publication of the paper by Qin et al. (2014) describing the Percentage of Conserved Proteins (POCP) to delineate genus. The number of valid genera is highlighted ten years later. The data was accessed on 2024-12-11 at the List of Prokaryotic names with Standing in Nomenclature (Parte et al., 2020).

Supplemental Information 5 Adequacy between POCP values computed with the reference BLASTP against the BLASTPDB method that build databases before alignment

Each point (n = 70,602) represents a POCP value between two genomes (see (1)). The colors represent the number of data points binned together in hexagons to avoid over-plotting. Coefficient of determination (R 2) and associated p-value are shown on top of each linear regressions.

Supplemental Information 6 Adequacy between POCPu values computed with the reference BLASTP against the BLASTPDB approach that build databases before alignment

Each point (n = 70,602) represents a POCPu value between two genomes (see (2)). The colors represent the number of data points binned together in hexagons to avoid over-plotting. Coefficient of determination (R 2) and associated p-value are shown on top of each linear regressions.

Supplemental Information 7 Distributions of POCPu values as in Fig. 4 broken down per bacterial family

The true category is based on the GTDB taxonomy. The family-specific POCPu thresholds for genus delineation proposed in this study were taken from Table 2 and are indicated with plain vertical line, whilst the default POCPu threshold of 50% is indicated by dashed lines.

Supplemental Information 8 Lack of association between genome or proteome size and POCPu

Distributions of differences in genome size (A) and proteome size (B) for families using default threshold or optimized thresholds. In case of an association between genome (or proteome) size and POCPu, we expected families for which optimized thresholds are proposed to have a shift towards larger differences explaining poor delineation performance in Fig. 5B. This was not the case, indicating that genome size and proteome size did not influence genus delineation. POCPu thresholds type were taken from Table 2.

The authors are grateful to Christian Schudoma, Daniel Podlesny, and Shahriyar Mahdi Robbani (European Molecular Biology Laboratory, Molecular Systems Biology Unit) for their help in fixing issues with an earlier version of the Nextflow workflow. Charlie Pauvert thanks all members of the Clavel Lab for constructive feedback on the figures at the lab retreat 2024.

Additional Information and Declarations

Competing Interests

Author Contributions

Data Availability

The authors declare there are no competing interests.

Charlie Pauvert conceived and designed the experiments, performed the experiments, analyzed the data, prepared figures and/or tables, authored or reviewed drafts of the article, and approved the final draft.

Thomas C.A. Hitch conceived and designed the experiments, analyzed the data, authored or reviewed drafts of the article, and approved the final draft.

Thomas Clavel analyzed the data, authored or reviewed drafts of the article, acquired funding, resources, and approved the final draft.

The following information was supplied regarding data availability:

All the proteins sequences used in the analysis were downloaded from the GTDB (r214) and are available at Zenodo:

– Pauvert C, Hitch, TCA, & Clavel T. (2025). Protein sequences from representative genomes of the Genome Taxonomy DataBase (r214) and associated metadata table (Version r214) [Data set]. Zenodo. https://doi.org/10.5281/zenodo.17286354.

The list of valid bacteria names is available at GitHub and Zenodo:

– https://github.com/thh32/Protologger/blob/0731adf80f1bbc5f8ee1904c8e9648ef45b13303/DSMZ-latest.tab.

– Thomas Hitch, & tclavel. (2025). thh32/Protologger: V1.1 (1.1). Zenodo. https://doi.org/10.5281/zenodo.17240891.

The raw output files are available at Zenodo: Pauvert C, Hitch TCA & Clavel T. (2025). pocpbenchmark: raw output files from benchmark workflow before analyses [Data set]. Zenodo. https://doi.org/10.5281/zenodo.14974869.

The cleaned and formatted POCP/POCPu data is available at Zenodo: Pauvert C, Hitch TCA & Clavel T. (2025). pocpbenchmark: POCP and POCPu values and metadata tables for analysis [Data set]. Zenodo. https://doi.org/10.5281/zenodo.14975029.

The Nextflow workflow used for the benchmark is available at GitHub and Zenodo:

– https://github.com/ClavelLab/pocpbenchmark/tree/v1.1.

– Pauvert, C. (2025). Nextflow workflow for benchmarking proteins alignment tools for improved genus delineation using the Percentage Of Conserved Proteins (POCP) (v1.1). Zenodo. https://doi.org/10.5281/zenodo.16688238.

The code is available at GitHub and Zenodo:

– https://github.com/ClavelLab/pocpbenchmark_manuscript/tree/v1.1.

– Pauvert, C. (2025). Code and manuscript for benchmarking proteins alignment tools for improved genus delineation using the Percentage Of Conserved Proteins (POCP) (v1.1). Zenodo. https://doi.org/10.5281/zenodo.17106830.

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
