# Peer review of "Fast and robust estimate of bacterial genus novelty using the percentage of conserved proteins with unique matches (POCPu)"

_PeerJ, doi:10.7717/peerj.20259_

## Round 0.1 · original submission · Major Revisions

· Academic Editor

Major Revisions

Please revise the manuscript accordingly based on the reviewers' comments.

Reviewer 1 ·

Basic reporting

The quality of the English is good. It is clear and ambigous, although there were a few minor issues.

The use of literature is fine, providing a good and up to date background context to the article.

The links to github etc all appear to valid with comprehensive material available via those links. Figures and tables are fine. The article is self-contained and clear.

There are some issues with the structure and presentation.

The authors define their aims as "(i) find the most suitable method to identify homologous protein sequences, (ii) eliminate the impact of duplicate genes on POCP calculations, and (iii) evaluate how POCP performs on a large set of known taxa."

These aims are fine enough, but the structure and section / subsection headings of the paper do not follow these aims too closely, making it harder to follow than it needs to be. As mentioned above, aim (ii) is introduced in the results section, rather than the methods. Arguably aim (iii) also.

In general, the section headings of Methods and Results are not well adhered to. Various chunks of text in the results would be better in the methods section. And the workflow could be more clear - a number of different analyses are applied but this is not clear from the methods section. Also, the POCPu approach is presented in the results section, rather than the methods. Likewise the data sets analysed are presented in the results, rather than the methods.

Experimental design

The experimental design is basically fine, but a lot of space is devoted to the speed-ups from the use of DIAMOND instead of BLASTP. This is not new knowledge, similar comparisons have been performed elsewhere (including papers cited by the authors) and in general the results are well known. It does not need to repeated here.

The novelty of this paper could therefore be better defined. The main novelty seems to be the suggestion for the POCPu method, which only uses unique matches, rather than the speed-ups derived from the use of DIAMOND instead of BLASTp.

As mentioned above, the actual workflow could be better defined and explained. Perhaps some material from the github pages could be reproduced here, or a figure introduced to summarise the analyses.

Validity of the findings

The results presented by the authors appear to be rigorous and robust. They demonstrate that their modification to the POCP method, POCPu, can improve bacterial genus delineation.

They might say more about the software they have made available and how that could be used by the community. There is remarkably little about these resources in the main body of the paper.

Additional comments

Other comments:

Line 24: 20x faster : you should explain what you are comparing to. Faster than what?

The list of genomes processed is provided in Table S1, which is helpful.

Line 134 Benchmarking methods for protein sequence alignment - perhaps this should more clearly state what principle features the authors are interested in. E.g. is it aim (i) i.e. the need to identify homologs or to obtain the POCP scores, rather than the more general problem of alignment. Or speed which has been explored elsewhere. Or maybe the interest is in a combination of the features.

A figure of the workflow would be helpful. It would help to visualise the process of how the work was actually done, given there are a number of stages and different analyses.

Results - Finding a BLASTP alternative for POCP. If this is the results for the benchmarking, it would help the reader if the terminology was consistent. Some of this material would be better in the methods e.g. lines 195-198 describing the taxa analysed.

Perhaps the proposal for POCPu would also be better in the methods section than the results. The logic for including it in the "Finding a BLASTP alternative for POCP" section is not clear. Is it a BLASTP alternative?

What exactly is a unique match? Does it ignore all paralogs? For clarity, this should be explicitly stated so that others can implement the algorithm.

Line 143: "POCPu is closer to BLASTP than POCP" this is poorly expressed and currently makes no sense. POCPu is not an alignment method. Perhaps you mean "POCPu using DIAMOND …" Or the way it is stated in lines 254 to 256. This part, up to line 259 seems to be the key point for the novelty of this paper, rather than speed improvements which have already been established.

line 249 "were way faster" perhaps a bit casual. Perhaps "Thankfully," on line 274 also.

lines 264 to 267 contain material more suitable for methods.

Lines 339-340: POPCu improves sustainability - but in comparison to what? Surely it is much the same as POCP given the same alignments must be performed? Or do you mean due to the use of DIAMOND - yet other methods cited in this paper also use DIAMOND.

Reviewer 2 ·

Basic reporting

no comment

Experimental design

no comment

Validity of the findings

no comment

Additional comments

The manuscript presents an updated tool POCPu designed for bacterial genus delineation using whole genomes, providing a robust framework that accurately and quickly classify organisms at the genus level. It would be a promising tool in the systematic microbiology. I think this work is well-presented with the clear structure, and the figures are also successful in illustrating the results. I have only several comments to improve the manuscript.

1. Code Accessibility: the source code is not available now, please host it on a public repository (e.g., GitHub) to enhance transparency and community contributions.
2. The authors tested thousands of genome data within a large scale of genera to assay POCP and POCPu tools. It looks like POCPu is much better than POCP, whatever for accuracy or speed. I am curious whether both tools have different preferences targeting different taxonomic groups?
3. The accuracy of 50% cut-off value showed high dependence on MCC, however, the authors did not define the MCC threshold values of high, middle and low. The genus with low MCC values does not appear suitable for POCPu; please list this limitation clearly.
4. References:
1). Lines 408, 451 and others: please use the Italic words of bacterial names.
2). Line 429: Bacteria and Archaea should be in upright letters.
3). The name of the Journals should be in consistence, abbreviation or full writing?
4). The title of articles should be in consistence, should the first letter of a word be capitalized or lowercase?

---

## Round 0.2 · accepted · Accept

· Academic Editor

Accept

The manuscript can be accepted now.

Reviewer 1 ·

Basic reporting

Basic reporting has been improved, with a clearer article structure and more references.


Line 99: “to” missing: “need a fast and reproducible framework TO classify genera that needs to be tested at a large scale."

Experimental design

No comment.

Validity of the findings

No comment.

Additional comments

The authors have done a good job addressing previous comments and the article seems to be much improved and suitable for publication.

Reviewer 2 ·

Basic reporting

The authors have sufficiently addressed the comments from the previous review and the manuscript is much improved. I have no comments here.

Experimental design

no comment

Validity of the findings

no comment

Additional comments

no comment